# Exercise Suppresses Head and Neck Squamous Cell Carcinoma Growth via Oncostatin M

**DOI:** 10.3390/cancers16061187

**Published:** 2024-03-18

**Authors:** Takuya Yoshimura, Yuka Hirano, Taiji Hamada, Seiya Yokoyama, Hajime Suzuki, Hirotaka Takayama, Hirono Migita, Takayuki Ishida, Yasunori Nakamura, Masahiro Ohsawa, Akihiro Asakawa, Kiyohide Ishihata, Akihide Tanimoto

**Affiliations:** 1Department of Oral and Maxillofacial Surgery, Kagoshima University Graduate School of Medical and Dental Sciences, Kagoshima 890-8520, Japanh-tkym@d1.dent.kagoshima-u.ac.jp (H.T.);; 2Department of Pathology, Kagoshima University Graduate School of Medical and Dental Sciences, Kagoshima 890-8520, Japanakit09@m3.kufm.kagoshima-u.ac.jp (A.T.); 3Department of Oral Surgery, Kagoshima Medical Center, National Hospital Organization, Kagoshima 892-0853, Japan; 4Faculty of Pharma-Science, Laboratory of Pharmacology & Neuroscience, Teikyo University, Tokyo 252-0176, Japan; oosawa.masahiro.bg@teikyo-u.ac.jp; 5Department of Psychosomatic Internal Medicine, Kagoshima University Graduate School of Medical and Dental Sciences, Kagoshima 890-8520, Japan; asakawa@m2.kufm.kagoshima-u.ac.jp

**Keywords:** exercise, myokine, head and neck squamous cell carcinoma, oncostatin M, sarcopenia

## Abstract

**Simple Summary:**

In recent years, much research has focused on how exercise improves cancer prognosis and how the protein myokine, which is associated with exercise, inhibits cancer progression. However, there have been few detailed studies on oral cancer. This study is the first to show that exercise inhibits the progression of oral cancer. While other studies have shown that exercise suppresses growth, decreases the tumor formation rate, or improves survival in patients with other organ cancers, this study showed that exercise inhibits tumor formation and growth in oral cancer and prolongs survival. Furthermore, while many studies have demonstrated an indirect immune-mediated mechanism by which exercise suppresses cancer, this study suggests the additional possibility that myokines released by exercise directly affect oral cancer cells. This study has the above novelties.

**Abstract:**

Major advances have been made in cancer treatment, but the prognosis for elderly cancer patients with sarcopenia and frailty remains poor. Myokines, which are thought to exert preventive effects against sarcopenia, have been reported to be associated with the prognosis of various cancers, but their effect on head and neck squamous cell carcinoma (HNSCC) is unknown. The aim of this study was to clarify the influence of exercise on the control of HNSCC and to examine the underlying mechanism involved. Mice were injected with HSC-3-M3 cells, a human cell line of highly metastatic and poorly differentiated tongue cancer, at the beginning of the study. Just prior to transplantation, blood was collected from the mice, and the levels of myokines were measured by ELISA. Oncostatin M (OSM), a selected myokine, was added to HSC-3-M3 cells, after which the cell proliferation ability, cell cycle, and protein expression were analyzed in vitro. Tumor cell viability was lower (control: 100%, exercise: 75%), tumors were smaller (control: 26.2 mm^3^, exercise: 6.4 mm^3^), and survival was longer in the exercise group than in the control group in vivo. OSM inhibited HSC-3-M3 cell proliferation in a concentration-dependent manner in vitro. The addition of OSM increased the proportion of cells in the G0/G1 phase, decreased the proportion of cells in the G2/M phase, and increased the expression of the CDK inhibitors p21 and p27. These results indicate that exercise may directly inhibit the proliferation of HNSCC cell lines via OSM.

## 1. Introduction

Great progress has been made in cancer treatment, but older patients and those with a variety of diseases are more difficult to treat [1]. Even within the same generation, treatment choices vary depending on the patient’s general condition. In recent years, patients have been conceptually categorized into fit patients who can undergo surgery and unfit patients who cannot undergo surgery; unfit patients often do not receive standard treatment [1,2,3]. Transforming unfit patients into fit patients is important in cancer treatment, and improving conditions such as frailty and sarcopenia due to lack of exercise can help this transformation [1,2,3].

Sarcopenia is characterized by an age-related loss of skeletal muscle mass, strength, and physical capacity [4,5,6]. The relationship between sarcopenia and the prognosis of various diseases has been studied. Sarcopenia has been noted to be a poor prognostic factor in cancers such as breast and colon cancer [4,7,8,9,10]. The quality and quantity of preoperative skeletal muscle in HNSCC patients reportedly affect patient prognosis [11,12,13]. However, the detailed mechanisms by which sarcopenia affects the prognosis of head and neck squamous cell carcinoma (HNSCC) patients have not been determined.

Exercise therapy is the most effective treatment for sarcopenia [14] and has been reported to inhibit the progression of various cancers and improve patient prognosis [15,16]. Exercise has previously been reported to mobilize immune cells and to be involved in suppressing cancer progression through the immune system [15,17,18]. However, few reports have shown that exercise and muscle contraction are directly related to cancer suppression without immune mediation [19], and the underlying mechanism remains unclear.

The cause of sarcopenia is not clear and is due to a complex interplay between various molecular mechanisms. In particular, the relationship between myokines, proteins produced by skeletal muscle fibers, and sarcopenia has been widely studied [20,21]. Myokine is an endocrine that is secreted from skeletal muscle upon exercise stimulation and is believed to exert systemic effects [20,21]. Myokines have been shown to exert anti-inflammatory and preventive effects against low-grade systemic inflammation such as age-related sarcopenia [21] and to be associated with prognosis in various cancers [22,23,24]. Myokines exert a paracrine/endocrine regulatory function on distant organs and tissues, as well as an autocrine function in regulating muscle metabolism [21,25,26]. Myokine levels are altered by exercise, and secreted myokines exert a direct inhibitory effect on proliferation by improving cell metabolism in remote organs through the bloodstream; they also play an important role in enhancing tumor cytotoxicity and immune cell infiltration [23]. Therefore, elucidating the function of myokines may lead to the development of new therapeutic strategies for patients with sarcopenia and cancer. Previously, exercise-induced upregulation of myokines was reported to inhibit the growth of prostate cancer cells [22]. The list of myokines is constantly growing, with hundreds of proteins now classified as myokines [25,26,27]. Oncostatin M (OSM), a myokine and a member of the IL-6 family, has been reported to be associated with the inhibition of cancer progression, especially in breast and lung cancers [28,29,30]. The associations of many myokines, including OSM, with cancer have been extensively studied in various cancer types [22,23,24,31]. There have been few reports on the associations between myokines and HNSCC, and it is unclear how myokines affect HNSCC [28,32]. 

The purpose of this study was to determine whether exercise inhibits the progression of HNSCC and to elucidate the mechanism of this inhibition.

## 2. Materials and Methods

### 2.1. Cell Lines and Culture

The HSC-3 and HSC-3-M3 cell lines were obtained from the Japanese Collection of Research Bioresources (JCRB) Cell Bank (Osaka, Japan), and the OSC-19 cell line was obtained from the RIKEN Bioresource Center (Tsukuba, Ibaraki, Japan). HSC-3 and HSC-3-M3 cells were maintained in minimum essential medium (MEM; Thermo Fisher Scientific, Tokyo, Japan) supplemented with 10% fetal calf serum. OSC-19 cells were maintained in a 1-to-1 mixture of Dulbecco’s modified Eagle’s medium with F-12 medium. All cell lines were maintained under standard conditions (37 °C in 5% CO_2_).

### 2.2. Mouse Models

All the studies complied with the ARRIVE guidelines and were approved by the Animal Experiment Committee of Kagoshima University (Approval No. D20005). The study design is shown in Figure 1. Four-week-old male BALB/C nu/nu mice (CLEA Japan, Inc., Tokyo, Japan) were placed in standard housing cages and maintained in a thermostatic environment under a 12 h light/dark cycle with free access to drinking water and food. All mice underwent a 1-week acclimation period prior to testing. For all the studies, the mice were randomized into groups with cages containing running wheels as a model of voluntary exercise (*n* = 20) or cages with no running wheel (*n* = 20) as a control at 4 weeks of age. The exercising mice had access to running wheels throughout the study. Mice were housed with only one in each cage. The total running distance was evaluated by a rotational frequency-measuring instrument on the running wheels. Body weight, wheel running distance, and food intake were measured twice weekly throughout the study period. Four weeks after study initiation, all the mice received tongue injections of HSC-3-M3 cells as follows. The HSC-3-M3 cells were cultured with 3 × 10^5^ cells/10 cm dish for 3 days, and cells that were approximately 80% confluent were collected and used. Mice were intraperitoneally anesthetized with a mixture of three anesthetics (0.3 mg/0.3 mL/kg dexmedetomidine, 4 mg/0.8 mL/kg midazolam, 5 mg/1 mL/kg butorphanol, and 2.9 mg/kg saline; adjusted to 0.05 mL per 10 g body weight), and HSC-3-M3 cells (1 × 10^5^/50 µL in HBSS) were injected into the tongue using a 27G Myjector. Tumor volume, tumor formation rate, wheel running distance, weight, food intake, and survival rate were measured twice weekly after HSC-3-M3 cell injection. Tumor volume was measured using a caliper with the tongue extended under intraperitoneal administration and calculated as tumor long diameter (a), short diameter (b), and thickness (c), using the following formula: V = 4/3 × π × a × b × c [33]. After measurements, the mice were intraperitoneally administered a medetomidine antagonist (antisedan 3.0 mg/0.6 mL/kg, phosphate-buffered saline 4.4 mL/kg, adjusted to 0.05 mL per 10 g body weight) to wake up the mice. Mice were checked twice daily and euthanized by cervical dislocation by skilled staff if they showed obvious abnormal behavior, such as weight loss of more than 20%, morbid physical findings, immobility, or tremors. Otherwise, the animals were euthanized 7 weeks after the start of the study. Blood was collected from 500 µL to 1 mL of blood from the orbital venous plexus of the mice immediately before tumor injection and at the time of euthanization.

### 2.3. ELISA

Mouse blood was centrifuged (4 °C, 13,000 rpm, 5 min) immediately after collection, and only the serum was transferred to 1.5 mL microtubes and stored at −80 °C until the start of the experiment. A Quantikine Mouse Oncostatin M ELISA Kit (R&D Systems, Inc., Minneapolis, MN, USA) was used to mix 50 μL of reagent and serum prepared according to the manufacturer’s protocol, which was subsequently added to the provided microplate. The absorbance of the standard was compared to the absorbance of the test sample to quantify the concentration of OSM. At least 100 μL of serum was obtained from the samples analyzed after centrifugation (*n* = 3).

### 2.4. Cell Viability and Proliferation

Cells were seeded overnight at a density of 2500 cells/well (96-well plate, Nalge Nunc, Thermo Fisher Scientific, Tokyo, Japan), treated or not treated with 100 ng/mL, and cultured for 2 days. The percentage of total cells in the Control and OSM groups on day 3 is shown with the Control group as 100%. HSC-3-M3 Cells were seeded overnight at a density of 2500 cells/well (96-well plate, Nalge Nunc, Thermo Fisher Scientific, Tokyo, Japan), treated or not treated with 1 ng/mL to 1000 ng/mL OSM, and cultured for 48 h. The number of viable cells was quantified immediately before the addition of OSM, 24 h after the addition of OSM, and 48 h after the addition of OSM. The number of viable cells was quantified using a Cell Counting Kit-8 (DOJINDO, Kumamoto, Japan) according to the manufacturer’s protocol. Cell viability was determined by measuring a generated formazan dye at 450 nm against a reference at 620 nm using a microplate reader. 

### 2.5. Cell Cycle Analysis

HSC-3-M3 cells were seeded overnight at a density of 3 × 10^5^ cells/10 cm dish (Nunclon Delta Surface, Thermo Fisher Scientific, Tokyo, Japan), treated or not treated with 100 ng/mL OSM, and were cultured for 3 days, after which the cells were collected. The recovered cells were fixed in 70% ethanol and stored at −20 °C until the start of the experiment. Fixed cells were added to 0.5 mL of FxCycle TM PI/RNase Solution (Thermo Fisher Scientific, Tokyo, Japan), and the cells were collected on an Invitrogen™ Attune™ NxT cytometer (Thermo Fisher Scientific, Tokyo, Japan) using a 488 nm excitation laser with a 574/26 nm absorption filter. The cell counts of PI-stained cells in the collected cell population were automatically analyzed by flow cytometer by DNA content, and histograms were generated. In addition, the flow cytometer analyzed the number of cells in the sharpest peak in the histogram (G0/G1 phase), the second-sharpest peak (G2/M phase), and the flat area between the peaks (s phase), and displayed these as a percentage of the total cell count.

### 2.6. Western Blotting

HSC-3-M3 cells were seeded overnight at a density of 3 × 10^5^ cells/10 cm dish (Nunclon Delta Surface, Thermo Fisher Scientific, Tokyo, Japan), treated or not treated with 100 ng/mL OSM, and were cultured for 3 days and then collected. Proteins were extracted with a RIPA Lysis Kit (ATTO, Tokyo, Japan). Cell lysates adjusted to equal protein concentrations were electrophoretically separated on polyacrylamide gels and transferred onto polyvinylidene difluoride (PDVF) membranes (Amersham Bioscience Health Care Bio-Sciences Corp., Piscataway, NJ, USA). The membranes were then incubated with primary antibody (1:1000 dilution) overnight at 4 °C, followed by incubation with anti-rabbit secondary antibody and anti-mouse secondary antibody (1:2000 dilution) for 1 h at room temperature. Then, an Odyssey CLx (SCRUM, Tokyo, Japan) was used to visualize specific protein bands by fluorescence luminescence. Measurements were quantified by dividing the fluorescence intensity of p21 and p27 by the fluorescence intensity of ACTβ. Primary antibodies [anti-p21 (DCS60, mouse mAb, #2946), anti-p27 (D69C12, rabbit mAb, #3686), and anti-β-actin (13E5, rabbit mAb, #4970)] were obtained from Cell Signaling Technology (Tokyo, Japan). Secondary antibodies [Anti-mouse IgG (H+L) (DyLight™ 680 Conjugate) #5470, Anti-rabbit IgG (H+L) (DyLight™ 680 Conjugate) #5366, Anti-rabbit IgG (H+L) (DyLight™ 800 4X PEG Conjugate) #5151, and Anti-mouse IgG (H+L) (DyLight™ 800 4X PEG Conjugate) #5257] were obtained from Cell Signaling Technology (Tokyo, Japan). 

### 2.7. Statistical Analysis

The results were analyzed with an unpaired *t* test using GraphPad Prism 9.5.1 software. The results are shown as the mean ± SD (for in vitro experiments) or mean ± SEM (for in vivo experiments), and differences for which *p* < 0.05 were considered significant. A log-rank test was performed for survival analysis, and differences of *p* < 0.05 were considered significant.

## 3. Results

### 3.1. Effects of Exercise on HNSCC-Bearing Mice In Vivo

#### 3.1.1. Changes in Body Weight, Food Intake, and Wheel Running Distance

There was no significant difference in body weight between the exercise group and the control group prior to injection (Figure 2a). Food intake was significantly greater in the exercise group throughout the entire period (Figure 2b). The wheel running distance increased before tumor injection but decreased gradually after injection (Figure 2c).

#### 3.1.2. Changes in Wheel Running Distance and Tumor Volume

Tumor formation was significantly lower in the exercise group than in the control group (Table 1). The mice in the exercise group that did not develop tumors had significantly longer average daily wheel running distances before tumor injection than tumor-bearing mice (Figure 3a). There was a weak negative correlation between average daily distance traveled before tumor injection and tumor volume (Figure 3b).

#### 3.1.3. Exercise Significantly Suppressed Tumor Growth and Prolonged Survival

Tumor size was significantly smaller in the exercise group than in the control group at all time points after tumor cell injection (Figure 3c). The overall survival rate in the exercise group was more than 70%, even 28 days after tumor seeding, whereas the overall survival rate in the control group was 50% at approximately 20 days after tumor seeding (Figure 3d).

#### 3.1.4. Exercise Increases OSM Levels

The OSM concentration in mouse blood in the exercise group (9.22 ± 0.66 pg/mL) was significantly greater than that in the control group (4.01 ± 2.11 pg/mL) (*p* = 0.03) (Figure 3e).

### 3.2. Direct Impact of OSM on HNSCC

#### 3.2.1. OSM Inhibits Cancer Cell Growth

All the cell lines supplemented with OSM exhibited significant inhibition of cell proliferation (Figure 4a). The ratios of the number of proliferating cells in the control group to that in the OSM 1 ng/mL group were 2.525 ± 0.14 and 2.243 ± 0.15 (*p* = 0.01) 24 h after addition of OSM and 5.428 ± 0.40 and 4.759 ± 0.34 (*p* = 0.01) 48 h after addition of OSM. The ratios of the number of proliferating cells in the control group to that in the 10 ng/mL OSM group were 2.233 ± 0.21 (*p* = 0.01) 24 h after addition of OSM and 4.575 ± 0.21 (*p* < 0.01) 48 h after addition of OSM. The ratios of the number of proliferating cells in the control group to that in the 100 ng/mL OSM group were 2.073 ± 0.23 (*p* < 0.01) 24 h after addition of OSM and 4.132 ± 0.29 (*p* < 0.01) 48 h after addition of OSM. The ratios of the number of proliferating cells in the control group to that in the OSM 1000 ng/mL group were 1.998 ± 0.17 (*p* < 0.01) 24 h after addition of OSM and 3.780 ± 0.25 (*p* < 0.01) 48 h after addition of OSM. The addition of OSM to HSC-3-M3 cells inhibited cell proliferation in a concentration-dependent manner (Figure 4b).

#### 3.2.2. OSM Inhibits Cell Cycle Progression in HSC-3-M3 Cells

The cell cycle distribution is shown as a percentage of the total cell count; the percentages of cells in the sub-G1, G0/G1, S, and G2/M phases in the control and OSM groups are shown (Figure 5a,b, Table 2). Compared to those in the control group, significantly more cells in the G0/G1 phase (*p* < 0.01) and significantly fewer cells in the G2/M phase (*p* = 0.01) were observed in the OSM group (Figure 5a,b, Table 2).

#### 3.2.3. OSM Increases the Expression of the CDK Inhibitors p21 and p27 in HSC-3-M3 Cells

In the control group, the p21/ACTβ ratio was 0.011 ± 0.011, and the p27/ACTβ ratio was 0.31 ± 0.066. In the OSM group, the p21/ACTβ expression level was 0.018 ± 0.018, and the p27/ACTβ expression level was 0.51 ± 0.060. Compared to those in the control group, the expression of p21 and p27 was significantly greater in the OSM-treated group (*p* < 0.01, *p* = 0.03) (Figure 6, Appendix A).

## 4. Discussion

It has been reported that exercise decreases the risks of development and progression of certain types of cancer [15,34]. Epidemiologic data show that physical activity significantly reduces the risk of 13 cancers, including HNSCC [34]. Previous studies have shown that wheel running prior to cell inoculation reduces tumor growth in melanoma, lung cancer, hepatocellular carcinoma, and breast cancer [15,16,35,36]. Other studies have shown that anaerobic exercise in mice significantly reduces the incidence of lung tumors [16,37]. A study analyzing other predictors associated with recurrence and survival in breast cancer patients showed that exercise reduced recurrence rates compared to those in inactive patients [38]. To our knowledge, no studies have examined in detail the effects of exercise on the growth and tumor formation in oral cancer patients. Our data show for the first time that exercise inhibits both tumor growth and tumorigenesis and prolongs survival in an orthotopic xenograft model of tongue cancer.

Previous reports have shown that HNSCC patients who develop sarcopenia have a poor prognosis; however, the mechanism involved remains unclear [11,12,13]. Many previous reports on some cancer types have demonstrated an association with myokines as one of the mechanisms by which exercise suppresses cancer [22,23,24,31]. During acute exercise, stress hormone concentrations, and muscle-derived plasma concentrations of myokines in particular, are reported to increase dramatically [15,39]. Past reports have shown that, when prostate cancer patients perform regular supervised exercise, blood levels of myokines such as secreted protein acidic and rich in cysteine (SPARC) and OSM are elevated compared to those in the no-exercise group [22]. Other studies have shown that OSM blood levels in mice measured immediately after forced exercise are significantly elevated compared to those in the control group and decrease to pre-exercise levels in approximately 2 h [40]. The present results are consistent with these previous findings. Our study shows for the first time that exercise increases blood levels of OSM in an in vivo xenograft model. The present study could not definitively prove whether OSM in the blood was actually produced in the muscle. However, previous reports have shown that muscle myokine and blood myokine concentrations increase after exercise [41], and it was presumed in the present study that the OSM produced in muscle was released into the blood after exercise.

Exercise has been reported to have no effect on body weight changes in mice but does have some effect on food intake [34,42]. Previous reports have shown that during the early phase of exercise intervention, food intake in the no-exercise group increases, whereas, during the late phase of intervention, food intake in the exercise group increases [42]. In another report, exercise had no effect on either weight change or food intake [34]. The present study showed that exercise does not affect weight change, which is consistent with previous findings. The results of this study also indicate that exercise increases food intake. Previous studies have not provided a unified view of changes in food intake due to exercise interventions. However, the results of this study showed no difference in nutritional status between the control and exercise groups before xenografting since there was no difference in body weight between the control and exercise groups, despite the greater dietary intake in the exercise group. Therefore, we can exclude the possibility that nutritional status before xenografting affects tongue cancer growth and survival.

Exercise is known to be involved in suppressing cancer progression via the immune system [15,17,18], but there are few reports on the mechanism by which exercise suppresses cancer without immune intervention. Myokines, which are elevated by exercise in prostate cancer patients, have been reported to inhibit cell proliferation when applied directly to cell lines in vitro [22]. Previous reports have shown that, in chondrosarcoma cell lines, OSM directly promotes the expression of Cdk inhibitors, such as p21 and p27, which inhibit cancer growth by arresting the cell cycle [28,31]; however, there are no reports on this effect in HNSCC. Few studies have reported the relationship between HNSCC and OSM, and, contrary to our results, some reports suggest that elevated OSM expression may be associated with accelerated cell cycle progression in HNSCC [32]. This study suggested that exercise may be directly involved in the inhibition of HNSCC growth via OSM, and the results are consistent with previous findings, revealing for the first time that exercise directly inhibits HNSCC growth via myokine production.

There are several limitations to this study. (1) We used only one cell line in this research. As shown in Figure 4a, it has already been shown that the addition of OSM inhibits cell proliferation in two cell lines (HSC-3 and OSC-19), but the growth inhibition mechanism of these cell lines is under investigation, and the cell growth inhibition mechanism of these cell lines may differ from that of HSC-3-M3 because of their different cell lines. Therefore, we plan to further investigate HSC-3, OSC-19, and other head and neck cancer cell lines in more detail, both in vivo and in vitro, in future studies. (2) We only examined one myokine related to exercise. As shown in previous studies, hundreds of myokines have been identified [26]. Clinical studies have also shown that exercise programs increase blood levels of several myokines in the serum of cancer patients [22]. Although we were able to examine only OSM in this study, we must consider that multiple myokines interact in vivo. We do not believe that OSM is the sole cause of this effect, but is rather one of many myokines. Therefore, further studies are necessary to identify different cytokines. 

## 5. Conclusions

We found that exercise inhibits the proliferation of HSC-3-M3 cells and increases the blood level of OSM in organisms. Furthermore, OSM inhibits HSC-3-M3 cell proliferation by increasing the expression of p21 and p27 and by blocking the transition from the G0/G1 phase to the S phase of the cell cycle.

## Figures and Tables

**Figure 1 cancers-16-01187-f001:**
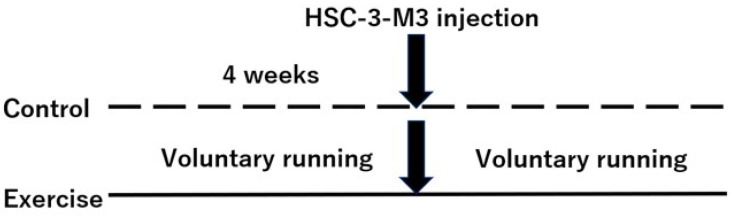
Study design. Mice were divided into an exercise group and a control group, and, four weeks after the start of the study, HSC-3-M3 oral cancer cells were transplanted. The mice were observed for another four weeks.

**Figure 2 cancers-16-01187-f002:**
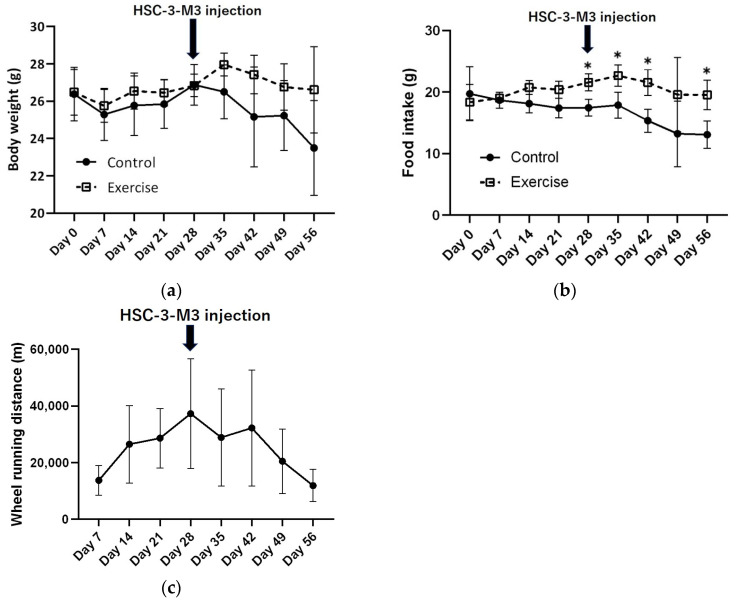
Changes in body weight and food intake of control and exercised group mice and wheel running distance of the exercised group mice in vivo. (**a**) Changes in body weight. (**b**) Changes in food intake. (**c**) Change in wheel running distance. *p* Values were calculated by *t* tests. * *p* < 0.05.

**Figure 3 cancers-16-01187-f003:**
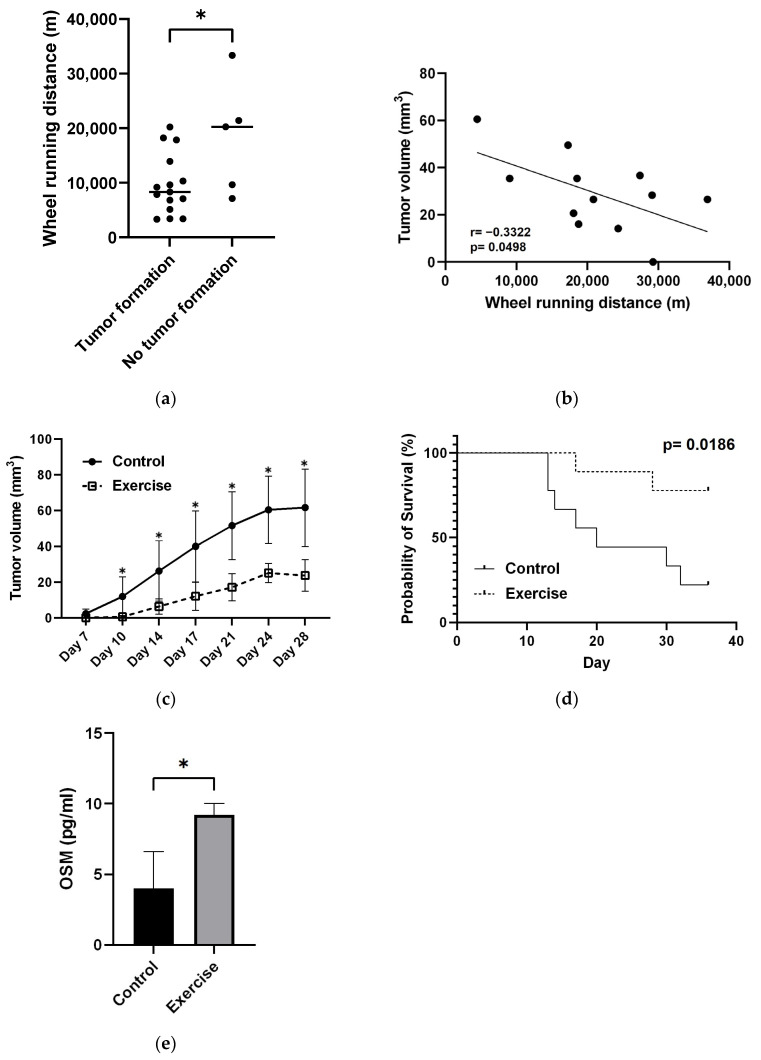
Effect of exercise on tumorigenesis and tumor volume, survival, and blood myokine levels. (**a**) Comparison of average daily wheel running distances before tumor injection in mice with and without tumor formation. * *p* < 0.05. (**b**) Correlation between tumor volume and average daily wheel running distances before tumor injection. (**c**) Changes in tumor volume after tumor transplantation in exercised and control mice. * *p* < 0.05. (**d**) Survival comparison between the exercised and control groups of mice. *p* values were calculated by the log-rank test. (**e**) Exercise-induced changes in blood oncostatin M (OSM) levels measured by ELISA. *p* values were calculated by *t* tests. * *p* < 0.05.

**Figure 4 cancers-16-01187-f004:**
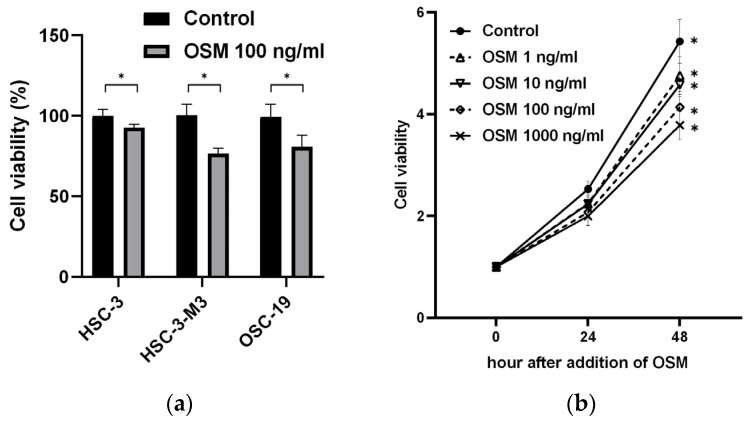
Effect of OSM addition on oral cancer cell lines. (**a**) Cell viability of OSM group against control group (100%) 48 h after addition of OSM using Cell Counting Kit-8. * *p* < 0.05. (**b**) Absorbance ratio of HSC-3-M3 treated with various concentrations of OSM using Cell Counting Kit-8 to that of OSM 24 and 48 h after addition of OSM. * *p* < 0.05.

**Figure 5 cancers-16-01187-f005:**
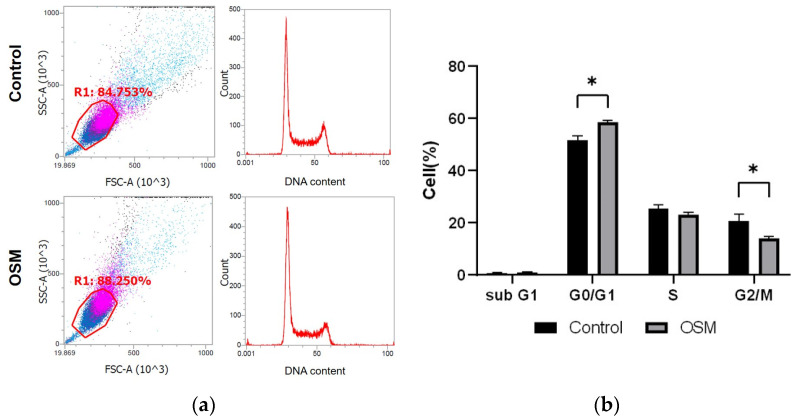
Effect of OSM addition on the cell cycle in HSC-3-M3 cells. Cell cycle was analyzed with Invitrogen™ Attune™ NxT cytometer by adding FxCycle TM PI/RNase Solution to each of the control and OSM groups. (**a**) The left figure shows the cell population analyzed by flow cytometry. Pink dots indicate a high cell population detected, while blue dots indicate a low cell population detected. The right figure shows the DNA content and cell count of the circled cell population. The y-axis indicates the number of cells and the x-axis indicates DNA content. (**b**) Comparison of the percentage of cells in each region analyzed by flow cytometer. * *p* < 0.05.

**Figure 6 cancers-16-01187-f006:**
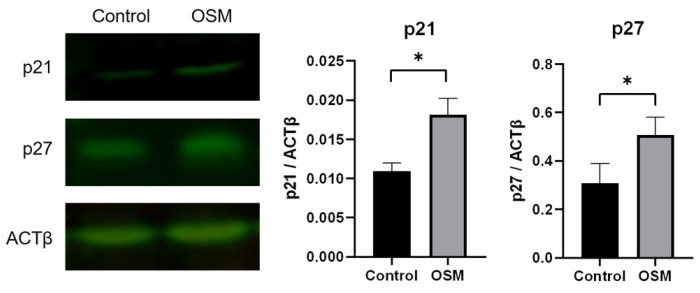
Effect of the addition of OSM on the expression of the CDK inhibitory factors p21 and p27 in HSC-3-M3 cells. Fluorescence Western blot results are shown for proteins from the control and OSM groups treated with p21 antibody (mouse mAb), p27 antibody (rabbit mAb), and anti-β-actin (rabbit mAb) and the corresponding secondary antibodies. These bands were read by Odyssey CLx, and fluorescence intensities were automatically measured. p21 and p27 fluorescence intensities against ACTβ fluorescence intensities are shown in the graphs. *p* Values were calculated by *t* tests. * *p* < 0.05, the original Western blots are shown in Appendix A. Source data in Appendix A.

**Table 1 cancers-16-01187-t001:** Comparison of tumor formation rates between mice in the control and exercised groups. The average tumor volume on day 14 after tumor transplantation is shown.

	Control	Exercise
Tumor formation rate	100% (20/20)	75% (15/20)
Tumor volume (mm^3^)	26.2 ± 16.9	6.4 ± 4.3

**Table 2 cancers-16-01187-t002:** Effect of OSM addition on the cell cycle in HSC-3-M3 cells. Numbers indicate the percentage of cells in each cell cycle.

	Control	OSM	*p* Value
Sub-G1	0.628333	0.892333	0.304222
G0/G1	51.74	58.68867	0.002743
S	25.42567	23.195	0.078736
G2/M	20.63833	13.94633	0.016104

## Data Availability

The data that support the findings of this study are available from the corresponding author.

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
