# Peer review of "Exercise Suppresses Head and Neck Squamous Cell Carcinoma Growth via Oncostatin M"

_cancers, 2024, doi:10.3390/cancers16061187_

Round 1
Reviewer 1 Report
Comments and Suggestions for Authors
Abstract
- It is important to distinguish which results were obtained in vivo and in vitro.
Introduction
- The paragraph from lines 55 to 62 is repeated in lines 63 to 69.
- On line 76, the authors stated that myokines are one of the causes of sarcopenia; I think it would be helpful to add an explanation as to how this is the case.
Materials and Methods
- I think the formula used to calculate tumor volume should be written out (line 120).
- On lines 125-126, it is written that antisedan was administered i.p. when mice were awake; should be written to wake up the mice.
- The number of cells/well as well as the types of plates used should be specified for all cell culture assays.
- The method used to observe cell viability and proliferation could be better described. I find it unclear when the absorbance was measured; was the absorbance measured in the same wells on different days (1, 2 & 3) ?
- Western Blot section is missing the information on the secondary antibodies : where were they purchased, catalogue numbers, indicate if are they coupled or not.
Results
Section 3.1.2.
- Since Table 1 is described before figure 3, I think it should be presented first.
- The references to figure 3 a and b are mixed up (lines 194-195).
- Line 194: “Tumor-free mice” refers to the mice in the exercise group which did not develop tumors?
Figure 3
- Letters are misplaced in the figure.
- Does “wheel running distance” represent the total running distance during the whole experiment or an average of running distance per day/per a certain period of time? Should be clarified.
- Unit of measurement in 3d (% on y-axis) is missing.
- The identification of the cytokine measured in 3e (OSM) is missing on the y-axis.
- The figure legend is lacking the method used to obtain the results presented (ELISA).
Section 3.2.1
- As mentioned in the materials and methods comments, it is unclear how this assay was conducted, therefore the way the ratios were calculated/their significance is also a bit unclear.
Figure 4
- Y- axis identification is missing (a and b).
- The figure legend is lacking the method used to obtain the results presented.
- I think it would be useful if the concentration of OSM used in 4a was included in the figure.
Figure 5
- The legend is missing.
- Maybe the figure should be separated with letters.
- I think it would be useful to describe how the percentage of cells in each stage of the cell cycle was measured. Perhaps some details could be added in the materials and methods section.
- The Western Blot results (p21/p27/ACTb) are presented in figure 5, then in figure 6.
Section 3.2.3
- I think it should be specified that p21 and p27 expression was measured in HSC-3-M3 cells.
Figure 6
- The legend is missing.
- Pictures are repeated from figure 5.
Discussion
- Line 309: the authors declared that OSM inhibits the growth of HSC-3 and OSC-9 cell lines, shouldn’t the HSC-3-M3 cell line also be mentioned?
Author Response
To Reviewer 1:
We would like to express our sincere appreciation for the important comments.
Abstract
#1 It is important to distinguish which results were obtained in vivo and in vitro.: We apologize for the insufficient explanation. We have added in vivo or in vitro to each of the results in abstract.(L35-40).
Introduction
#2 The paragraph from lines 55 to 62 is repeated in lines 63 to 69.: We apologize for this error, and we have deleted the text of lines55 to 62.
#3 On line 76, the authors stated that myokines are one of the causes of sarcopenia; I think it would be helpful to add an explanation as to how this is the case.: We appreciate your important suggestion. We have added an explanation of the relationship between myokine, sarcopenia and cancer according to the reviewer’s recommendation. (L69-82)
Materials and Methods
#4 I think the formula used to calculate tumor volume should be written out (line 120).: We apologize for the insufficient explanation. We have added the formula used to calculate tumor volume.(L122-125)
#5 On lines 125-126, it is written that antisedan was administered i.p. when mice were awake; should be written to wake up the mice.: We apologize for this error, and we have changed the description according to your point. (L125-128)
#6 The number of cells/well as well as the types of plates used should be specified for all cell culture assays.: We apologize for the insufficient explanation. We have added the description in the Materials and Methods (L148, L152, L160, L173).
#7 The method used to observe cell viability and proliferation could be better described. I find it unclear when the absorbance was measured; was the absorbance measured in the same wells on different days (1, 2 & 3) ?: We apologize for the insufficient explanation. The absorbance measurements were not taken in the same wells, but rather in several plates, each at a different time, as in the general method. We have modified the description in the Materials and Methods and Figure 4 (L148-155, L157-158).
#8 Western Blot section is missing the information on the secondary antibodies : where were they purchased, catalogue numbers, indicate if are they coupled or not.: We apologize for the insufficient explanation. We have added the information on the secondary antibodies.(L184-190)
Results
Section 3.1.2.
#9 Since Table 1 is described before figure 3, I think it should be presented first.: We sincerely agree with the reviewer’s suggestion. We have changed the Table 1 to the bottom of Section 3.1.2.(L212)
#10 The references to figure 3 a and b are mixed up (lines 194-195).: We apologize for this error, and we have changed the description. (L209-212)
#11 Line 194: “Tumor-free mice” refers to the mice in the exercise group which did not develop tumors?: We apologize for the insufficient explanation. You are right. It was a difficult description to understand, so we have changed the description to what you have indicated. (L209)
Figure 3
#12 - Letters are misplaced in the figure.: We apologize for this error. Corrected the position of the r and p values in Figure 3b, and changed the position of * in Figure 3d to the p value because it was misaligned. The other Letters were positioned in the optimal field by Prism software.
#13 Does “wheel running distance” represent the total running distance during the whole experiment or an average of running distance per day/per a certain period of time? Should be clarified.: We apologize for the insufficient explanation. “Wheel running distances” shows the average daily distance travelled before tumor injection. We added the description to clarify. (L209-212, L227-229)
#14 Unit of measurement in 3d (% on y-axis) is missing.: We apologize for the insufficient explanation. We have added the Unit of measurement in 3d (% on y-axis).
#15 The identification of the cytokine measured in 3e (OSM) is missing on the y-axis.: We apologize for the insufficient explanation. We have added the description of OSM.
#16 The figure legend is lacking the method used to obtain the results presented (ELISA).: We apologize for the insufficient explanation. We have added the description about ELISA. (L231-232)
Section 3.2.1
#17 As mentioned in the materials and methods comments, it is unclear how this assay was conducted, therefore the way the ratios were calculated/their significance is also a bit unclear.: We apologize for the insufficient explanation. As stated earlier, we have modified the description in the Materials and Methods and Figure 4 (L148-155, L157-158).
Figure 4
#18 Y- axis identification is missing (a and b).: We apologize for the insufficient explanation. We have added the Y-axis identification(a and b).
#19 The figure legend is lacking the method used to obtain the results presented.: We apologize for the insufficient explanation. We have added the description about method for figure legend.
#20 I think it would be useful if the concentration of OSM used in 4a was included in the figure.: We apologize for the insufficient explanation. We have added the description of the concentration of OSM.
Figure 5
#21 The legend is missing.: We apologize for this error. We have added the legend. (L258-263)
#22 Maybe the figure should be separated with letters.: We sincerely agree with the reviewer’s suggestion. We have separated the figure.
#23 I think it would be useful to describe how the percentage of cells in each stage of the cell cycle was measured. Perhaps some details could be added in the materials and methods section.: We apologize for the insufficient explanation. We have added the explanation according to the reviewer’s recommendation.(L166-171)
#24 The Western Blot results (p21/p27/ACTb) are presented in figure 5, then in figure 6.: We apologize for this error, and we have deleted.
Section 3.2.3
#25 I think it should be specified that p21 and p27 expression was measured in HSC-3-M3 cells.: We sincerely agree with the reviewer’s suggestion. We have added the description about HSC-3-M3.(L267-268)
#26 Figure 6 - The legend is missing.: We apologize for this error. We have added the legend.(L276-280)
#27 Pictures are repeated from figure 5.: We apologize for this error, and we have deleted.
Discussion
#28 Line 309: the authors declared that OSM inhibits the growth of HSC-3 and OSC-9 cell lines, shouldn’t the HSC-3-M3 cell line also be mentioned?: We apologize for the insufficient explanation. The limitation is described here. We have made changes to the description to make it clearer. (L339-347)
We look forward to hearing from you regarding our submission. We would be glad to respond to any further questions and comments that you may have.
Reviewer 2 Report
Comments and Suggestions for Authors
In this article authors investigated the effect of exercise on head and neck squamous cell carcinoma. Overall experiments were conducted well, and data is represented well.
Below are few suggestions:
What is rational for using 4-week-old male mice? Sarcopenia is for older patients; did authors consider age and gender of mice while performing this experiment. Because these kinds of experiments might affect with age and gender.
Please list engraftment rate between both group of mice. Is there an engraftment difference with exercise?
Please label OSM on figure 3e.
P21 and P27 data has been shown 2 times in figure 5 and figure 6. Please correct that.
Author Response
To Reviewer 2:
We would like to express our sincere appreciation for the important comments.
#1 What is rational for using 4-week-old male mice? Sarcopenia is for older patients; did authors consider age and gender of mice while performing this experiment. Because these kinds of experiments might affect with age and gender.: We sincerely agree with the reviewer’s suggestion. Although the age and sex of the mice should be taken into account in sarcopenia-related studies such as this study, other factors such as metabolic abnormalities and organ failure in old nude mice are likely to have a greater influence on the results, as this is a relatively long-term study. In previous similar studies(ref. 15), old mice and eight-week-old mice were used, and tumor growth in both mice was suppressed by exercise. Another similar study (Linda A. Buss, et al., 2020.) also used 6-10 week old mice. In addition, female mice are generally more prone to individual differences due to variations in their sexual cycle. Therefore, we decided to use 4-week-old male nude mice to eliminate another factor associated with acclimation period and aging. However, in view of your suggestion, we plan to conduct future experiments using syngeneic transplantation and carcinogenesis models in old mice instead of using nude mice. Thank you for your excellent suggestions.
#2 Please list engraftment rate between both group of mice. Is there an engraftment difference with exercise?: We apologize for the insufficient explanation. As shown in Table 1, the control group had a 100% engraftment rate, whereas the exercise group had a significantly lower engraftment rate of 75%.
#3 Please label OSM on figure 3e.: We apologize for the insufficient explanation. We have added the description of OSM on figure 3e.
#4 P21 and P27 data has been shown 2 times in figure 5 and figure 6. Please correct that.: We apologize for this error, and we have deleted.
We look forward to hearing from you regarding our submission. We would be glad to respond to any further questions and comments that you may have.
Reviewer 3 Report
Comments and Suggestions for Authors
The paper by Yoshimura et al examines the relationship between exercise and tumor growth and studies the underlying mechanism that connects the two. Overall, the paper contains some interesting data, but the scope of the study is rather limited. Additional data clarifying the relationship between exercise and tumor growth would strengthen the paper considerably. The paper is well written, but contains some obvious and important errors. A major revision with some additional data would make this a much more impactful paper.
Specific concerns are listed below.
Line 76, the first two sentences are not clearly stated and need to be revised. The paper states “Myokines, one of the causes of sarcopenia [19,20] , have been reported to be associated with the prognosis in various cancers [21-23]. Myokines are believed to exert systemic effects via autocrine, paracrine, or endocrine secretion via exercise [20].” The first sentence states that myokines causes sarcopenia, which is the opposite of the mechanism stated in the paper. The next sentence is also not clear about the relationship among myokines, exercise and the secretion of autocrine, endocrine and paracrine.
The data clearly establishes a correlation between exercise and tumor formation, as the exercise group had few mice develop tumor and their tumors are also smaller than the control group. The inverse correlation between the running distance and tumor size is also interesting, but the causal relationship is harder to decipher. It is not clear why some mice run more or less than others: Is it because they had a bigger (thus running less) or smaller tumor (thus running more) than others, or is it because some mice are healthier than others and thus ran more and had smaller tumors? In both scenarios, running distances are not the original variable and thus not the cause of the size of the tumors. It is possible that some mice just decided to run more than others, and the longer running distance caused the tumor to grow slower. These questions can be clarified by the distances run by each mouse before and after the cancer cell injection and by controlling the running distance, say by making the running wheels available for different times. Can the authors provide more information and discussion on these questions?
Oncostatin M could very well be the main link between exercise and tumor growth, but it cannot be concluded to be the only factor as the title of the paper suggests. Other myokines and mechanisms could very well contribute to the connection.
Another weakness of the paper is that this phenomenon is observed with one cell line. It is understandable that tumor models with more cell lines would expand the study considerably, nevertheless the question remains if the observed effects are also applicable to other cell lines and thus head and neck squamous cell carcinoma in general.
The first part of Fig 5 appears to be the same as Fig 6.
Comments on the Quality of English LanguageWell written except some needed revisions.
Author Response
To Reviewer 3:
We would like to express our sincere appreciation for the important comments.
#1 Line 76, the first two sentences are not clearly stated and need to be revised. The paper states “Myokines, one of the causes of sarcopenia [19,20] , have been reported to be associated with the prognosis in various cancers [21-23]. Myokines are believed to exert systemic effects via autocrine, paracrine, or endocrine secretion via exercise [20].” The first sentence states that myokines causes sarcopenia, which is the opposite of the mechanism stated in the paper. The next sentence is also not clear about the relationship among myokines, exercise and the secretion of autocrine, endocrine and paracrine.: We sincerely agree with the reviewer’s suggestion. The statement in line 76 was incorrect as you pointed out. Myokine is an endocrine substance secreted by skeletal muscle upon exercise stimulation and is thought to act systemically. And previous reports have shown that it exerts anti-inflammatory and prophylactic effects on low-grade systemic inflammation such as age-related sarcopenia, suggesting that it may contribute to the development of treatments for sarcopenia. We have reviewed the references again, revised the manuscript substantially, and changed the correct meaning. We have also added clarification on the relationship between myokine, exercise, autocrine, endocrine, and paracrine secretion. (L69-82)
#2 The data clearly establishes a correlation between exercise and tumor formation, as the exercise group had few mice develop tumor and their tumors are also smaller than the control group. The inverse correlation between the running distance and tumor size is also interesting, but the causal relationship is harder to decipher. It is not clear why some mice run more or less than others: Is it because they had a bigger (thus running less) or smaller tumor (thus running more) than others, or is it because some mice are healthier than others and thus ran more and had smaller tumors? In both scenarios, running distances are not the original variable and thus not the cause of the size of the tumors. It is possible that some mice just decided to run more than others, and the longer running distance caused the tumor to grow slower. These questions can be clarified by the distances run by each mouse before and after the cancer cell injection and by controlling the running distance, say by making the running wheels available for different times. Can the authors provide more information and discussion on these questions?: We sincerely agree with the reviewer’s suggestion and apologize for the insufficient explanation. As you point out, it is difficult to show a causal relationship between running distance and tumor size in this study. However, Figures 3a and 3b show the daily average of running distance before tumor injection. Therefore, they show the effect of average running distance before tumor injection on tumor formation and tumor growth, thus ruling out the possibility that tumor size affects running distance. During the acclimation period, the mice are assigned to have no differences in body weight, diet. We also believe that there are no individual differences in health status, as there are no noticeable differences in body weight and food intake before tumor injection, as shown in Fig. 2. However, the manuscript lacked a statement that the running distance shown in Figs. 3a and 3b represents the average daily distance before tumor injection, so we have added a statement in section 3.1.2. and in the legend of Fig. 3 (L219-212, L227-229). As in similar studies in the past, this study chose voluntary wheel running so as not to stress the mice. However, in light of your suggestion, we plan to conduct future studies using forced-exercise mice and controlling the running distance. Thank you for your excellent suggestions.
#3 Oncostatin M could very well be the main link between exercise and tumor growth, but it cannot be concluded to be the only factor as the title of the paper suggests. Other myokines and mechanisms could very well contribute to the connection.: We sincerely agree with the reviewer’s suggestion. As stated in line the limitation (L345-353), this study was able to examine OSM only, but we have to take into account that there are multiple myokines interacting in vivo. We do not consider OSM to be the only cause of this effect, but one of many myokines. As in previous studies, many myokines have been found to be altered, and studies on other myokines are underway, but many things are still under investigation, such as invasion and metastasis, and will be reported in future studies. Further studies are planned in the future, so we hope you will be able to confirm this there.
#4 Another weakness of the paper is that this phenomenon is observed with one cell line. It is understandable that tumor models with more cell lines would expand the study considerably, nevertheless the question remains if the observed effects are also applicable to other cell lines and thus head and neck squamous cell carcinoma in general.: We sincerely agree with the reviewer’s suggestion. As stated in the limitation the limitation (L339-345), this is a major issue in this paper and we believe that further research is needed. We are planning to conduct further studies using an exercise group, a frailty group and a control group, which will be discussed there with other head and neck cell carcinoma lines. Thank you for your excellent suggestions.
#5 The first part of Fig 5 appears to be the same as Fig 6.: We apologize for this error, and we have deleted.
We look forward to hearing from you regarding our submission. We would be glad to respond to any further questions and comments that you may have.
Round 2
Reviewer 3 Report
Comments and Suggestions for Authors
The revision addressed my concerns. This is an interesting and potentially important paper with some questions to be addressed in future studies.
Author Response
We appreciate your excellent suggestion.
We will take your suggestions into consideration to develop our future research.
Thank you very much.